# Pulsed Laser Spot Welding Thermal-Shock-Induced Microcracking of Inconel 718 Thin Sheet Alloy

**DOI:** 10.3390/ma16103775

**Published:** 2023-05-17

**Authors:** Mingli Shi, Xin Ye, Yuanhao Wang, Di Wu

**Affiliations:** 1School of Materials Engineering, Shanghai University of Engineering Science, Shanghai 201620, China; sml2022315@163.com (M.S.);; 2Shanghai Collaborative Innovation Center of Laser Advanced Manufacturing Technology, Shanghai 201620, China

**Keywords:** pulsed laser spot welding, thermal shock, heat-affected zone, microcrack

## Abstract

This paper investigates the change in solidification microcrack susceptibility under the influence of thermal-shock-induced effects for pulsed laser spot welding molten pools with different waveforms, powers, frequencies, and pulse widths. During the welding process, the temperature of the molten pool under the effect of thermal shock changes sharply, triggering pressure waves, creating cavities in the molten pool paste area, and forming crack sources during solidification. The microstructure near the cracks was analyzed using a SEM (scanning electron microscope) and EDS (electronic differential system), and it was found that bias precipitation occurred during the rapid solidification of the melt pool, and a large amount of Nb elements were enriched in the interdendritic and grain boundaries, which eventually formed a liquid film with a low melting point, known as a Laves phase. When cavities appear in the liquid film, the chance of crack source formation is further increased. Using a slow rise and slow fall waveform is good for reducing cracks; reducing the peak laser power to 1000 w is good for reducing cracks in the solder joint; increasing the pulse width to 20 ms reduces the degree of crack damage; reducing the pulse frequency to 10 hz reduces the degree of crack damage.

## 1. Introduction

Due to its quick welding cycle, large adjustment range, and high intensity, pulsed Nd:YAG (neodymium-doped yttrium aluminum garnet) laser welding is frequently employed to combine incompatible materials [1,2]. Pulsed laser spot welding is a new spot welding technology with high welding intensity, adaptability, and welding accuracy through the interaction of a single pulse laser beam with the metal [3,4], and has a wide range of applications, such as pressure sensors, pacemakers, and fuel fields, with high development prospects [5,6]. In addition, it is anticipated that in the future, pulsed laser spot welding will take the role of conventional contact spot welding. However, with laser, electron beam, and other high-energy beam welding processes, the temperature of the material in the high-energy beam concentrated bombardment area rises sharply, and melting and vaporization occurs. This rapid energy deposition causes the welded material to respond to thermal stresses with significant suddenness, resulting in strong shock loads that cause cracks and other damage to the material.

Inconel 718 alloy is used in a wide range of applications, such as chemical, petrochemical, energy, and aerospace. However, Inconel 718 alloy is prone to some metallurgical problems in fusion welding, especially Nb segregation and coarse reticulated Laves in the fusion zone (FZ) [7]. This significantly reduces the mechanical properties of the welded joint, and the liquefaction of the niobium-rich phase in the heat-affected zone (HAZ) also leads to microcracking problems [8]. In addition, it was found that the proximal stage of solidification within the solidification melt pool or paste zone is prone to solidification cracking where dendrites grow into complete grains. The solidified dendrites impede the flow of the remaining liquid, resulting in the possibility that the liquid metal may not be adequately supplied to the dendrite space. At the same time, solidification shrinkage and thermal contraction generate tensile stresses/strains that are concentrated near the liquid film region. When the local tensile stress/strain exceeds the crack resistance of the material, solidification cracking occurs [9]. Janaki Ram et al. [10] used welding current pulsation in GTAW of Inconel 718 and observed dendrite refinement in the FZ. Sivaprasad et al. [11] implemented arc oscillation in GTAW of Inconel 718 alloy and observed that dendrite refinement in the FZ led to the development of finer Laves phases in the FZ.

The formation of microcracks in the HAZ of welded joints is usually associated with shrinkage strain during solidification and structural liquefaction of Nb/C and Laves phases at grain boundaries [12]. During the fusion welding process, low melting point eutectic components form along the grain boundaries in the HAZ and the welding thermal cycle causes plastic strain. If the liquefied film cannot withstand the thermally induced strain during cooling, intergranular microcracks will form [13]. Liquefaction in the grain boundaries is due to the lower melting point of the Nb-rich, MC-type carbides and Laves phases heated above the eutectic temperature in the HAZ during welding [14]. The tendency of Inconel 718 welds to microcrack in the HAZ depends on grain size, post-weld heat treatment, base material (cast or forged), presence of trace alloying elements, and weld heat input [15]. Lin’s study concluded that with larger grain size, the possibility of weld intersection with the grain is reduced and, therefore, the possibility of microcracking is reduced [16]. Low [17] concluded that heat treatments that dissolve intermetallic components and eutectic phases would eliminate the possibility of liquefaction at lower temperatures and improve crack resistance. Han [18] concluded that thermal cracking occurs during the solidification of molten metal under laser heating and that the crack length can be reduced by decreasing the heat input. Keivanloo [19] found that reducing the cooling rate reduced residual stresses and decreased crack sensitivity.

The issue of thermal shock welding has received a great deal of attention, and the heat-affected zone cracking induced by electron beam thermal shock has been studied, as well as the exploration of thermal excitation caused by pulsed laser welding. In order to further study the pulsed laser spot welding melt pool and micro-cracking induced by thermal shock, this paper chose the rolled state Inconel 718 alloy sheet as the weld substrate and carried out spot welding experiments with different parameters. By analyzing the melt pool pressure, the pulsed laser thermal shock effect is studied, and the crack formation mechanism is elaborated. Analysis of cracking and precipitation phase composition in welded joints is performed using SEM and EDS to explore crack damage from pulsed laser thermal shock.

## 2. Experimental Procedure

### 2.1. Experimental Materials

The sample size is 200 mm × 100 mm × 1 mm, and the weld base material is the nickel-based alloy Inconel 718 (composition and mechanical characteristics are shown in Table 1 and Table 2). The microstructure of the base material is shown in Figure 1, which shows that the grains of the base material are equiaxed and the grain size is about 20 μm. From Figure 1, it can be observed that some black particles are diffusely distributed in the visible spectrum; these black particles are M_23_C_6_ carbides, which are basically distributed inside the crystal and rarely located at the grain boundaries. The parent substance also contains some lumpy MC carbides, but no Laves phase was discovered.

### 2.2. Experimental Procedure

The schematic diagram of the spot-welding experimental equipment is shown in Figure 2. Before beginning the welding experiment, sample material’s surface was cleaned with alcohol to remove any impurities that might influence the outcome of the test. Then, the sample material was set down on the welding bench and the test was begun.

The experimental Trupulse556 pulsed laser has a TEM00 mode, a spot diameter of 600 μm, a maximum power rating of 10 kw, a laser wavelength of 1064 nm, and a focal length of 232 mm. The start point, end point, and weld duration are established using the robot-controlled laser head from ABB (Asea Brown Boveri), which has a repeatable positioning accuracy of 0.01 mm for spot welding. The welding experiments are mainly pulsed laser spot welding using a Nd:YAG laser with a wavelength of 1064 nm. Pulsed laser conduction spot welding with different powers, frequencies, and pulse widths was performed under two waveforms.

The specimens were wire cut and made into samples containing the welded joint part after welding experiments for metallographic preparation and characterization, and polished after grinding with 1500# type sandpaper. Alcohol, hydrochloric acid, and copper chloride were prepared in accordance with the ratio of 100:100:5 to form a corrosion solution, the specimens were corroded for 10 s, and a light microscope was used to observe the surface of the welding spot. In order to study the variations of the size and number of cracks in welded joints with different laser parameters, the images of the welded joint surfaces were processed using imagej to count the size and number of cracks in the welding spot.

## 3. Results and Discussion

The shock wave induced by the heat conduction of pulsed laser welding has an impact effect on the interior of the welded joint, which may lead to different degrees of damage in the welded joint, such as segregation, microcracking, and porosity. In this chapter, cracks in pulsed laser spot welding are studied to analyze the composition of the cracks, the surrounding precipitated phases, and the effect of laser parameters on the crack.

### 3.1. Cracks in Welding Spot

Figure 3 shows an optical microscope image of a solder joint with some precipitated phases irregularly distributed in the joint with a size of 20–40 μm. There are cracks in the joint with a size of about 100 μm, and the cracks are connected to the precipitated phases.

The crack damage is less when the welding power is lower. When the power is higher, cracks are more likely to appear in the welded joints, and the cracking injury is more serious. If other parameters remain unchanged, the crack damage in the welded joint is lighter when using the slow rise and slow fall waveform; the crack damage in the welded joint is heavier when using the rectangular waveform. The results show that the welding process, waveform, and thermal cycling process are closely related.

#### Crack Formation Mechanism

Figure 4a–d show the surfaces of the welded joints after welding at different peak powers; Figure 4e–h show the corresponding cracks in the welded joints at different peak powers. It can be found that if the other parameters remain the same, the crack damage is lighter when the welding power is lower. When the power is higher, cracks are more likely to appear in the welded joint, and the crack injury damage is more serious. Changing the welding parameters cannot eliminate the crack in the welded joint, indicating that the thermal shock of the welding process causes cavities in the solidification process of the molten pool, and these cavities develop into cracks under the action of thermal stress.

Wang [20] stated that solidification cracks form in the paste region, which is the most vulnerable zone in the liquid-solid mixing region (0.9 < fs < 0.99). The paste zone is a two-phase zone between the all-liquid and all-solid states. In this region, all solidification microstructures and defects are formed. This region can be divided into three regions: (1) the easy liquid supply region, which lasts until the volume fraction of solids fs ≈ 0.9, (2) the vulnerable region with restricted feeding, and (3) the coherent region, where the remaining liquid is in the form of isolated droplets, i.e., no feeding is possible. In region 1, any strain can be compensated by liquid flow. In region 3, a coherent framework of solids containing a certain percentage of droplets is retained.

The latter two-phase solid can survive the shear pressures produced during the curing process and possesses strength that is approximately equal to that of a solid. The low volume fraction of liquid is present as a very thin film between the agglomerated dendrites during the solidification phase close to the paste zone in the critical region 2. Fluid supply is problematic in this area, and the material’s transverse shear strength is very low. As a result of this, the flow of liquid or solid may not be able to accommodate the development of strain brought on by thermal contraction and/or external deformation of the solid, and the dendrites will be forced apart, leading to hot ripping. As demonstrated in Figure 5, the latter two areas are constrained or unable to feed due to thermal shock caused by pulse laser welding in the paste area. The technique of pulse laser welding creates cavities and shock waves and pressure waves in the latter two zones, which serve as a source of cracking.

### 3.2. Precipitation Phase and Crack Composition in Welding Spot

#### 3.2.1. Precipitation Phase Composition in the Welding Spot

The SEM of the precipitated phase’s surface in the welded joint can be seen in Figure 6a, and Figure 6b shows the distribution concentration of each element. It was discovered that the measured elements have a very variable carbon element composition and a strong carbide precipitation at the crack. The formation of the MC phase here during the welding process results in a significant amount of segregation, which may be the cause of the increase in carbides. The data show that Nb, Ti, Al, and other elements are present in the precipitates. These findings suggest that as a result of thermal stress, component fusion precipitation caused by the enriched alloying elements occurs in the weld area, which is isolated by its high temperature. During the final step of solidification, the molten metal produces a precipitated phase on the fracture surface as the temperature drops.

The molten metal solidifies at the high temperature of the operation, and during this time, the Nb-rich Laves phase precipitates. More material with a low melting point solidifies in the last stage of the grain boundary if there is a significant amount of Laves phase precipitated close to the crack. Thermal cracking is more likely to happen at this point if the liquid components are linked and the boundary lengthens down the grain.

#### 3.2.2. Crack Composition in the Welding Spot

The welded joint contains solidification cracks, also known as thermal cracks, which develop during the solidification of the molten pool at the weld’s end. The development of the low melting point Laves phase is primarily responsible for the development of thermal fractures in welded seams during solidification. Due to the complexity of the base material’s alloy, rapid cooling is more likely to cause elemental segregation than solid solution treatment. Compounds and solid solutions can form in specific local locations if the elemental abundance is high enough. The temperature range for solidification widens if the melting point of the new phase is lower than that of the parent metal. As a result, the metal melts as the laser is at the edge of the sample. As the laser moves over the metal, the metal is rapidly cooled. Metals with a certain composition have the tendency to maintain their corresponding tissue composition according to the thermodynamic equilibrium conditions. Therefore, if a change in temperature occurs, it will cause a change in the organization; this change is controlled by the diffusion behavior. For rolled-state 718 joints, the liquid film solidifies during cooling, precipitating the Nb-rich Laves phase and the MC carbide phase. The solidification process involves redistribution of solute elements, and the precipitated phases are related to Ni, Nb, and C. The cracked surface in the welded joint was observed by SEM as shown in Figure 7a. The Nb content in the alloy increased; the alloy was analyzed by EDS on the crack surface, and the results are shown in Figure 7b. As shown in Figure 7c, the crack surface’s Nb content is much higher than the Nb content in the branch alloy. The Nb and C contents in the A region of the crack surface were examined by EDS, and statistical data were obtained from the crack surface separately. According to our interpretation, the Nb-rich interdendritic liquid film can precipitate NbC and Laves phase and ultimately remain on the crack surface because L + r does not undergo the Laves reaction due to the greater C/Nb ratio, creating more NbC and less Laves phase. This shows that the solidification of low melting point phases, such as Nb-rich Laves eutectic and carbide precipitates, is intimately related to solidification cracking.

Elemental segregation takes place during rapid solidification, and solid solutions and low melting point compounds are created close to the crack. Low melting points cause it to propagate into the molten metal along the Laves phase due to its limited ductility at high temperatures. The molten metal stays liquid at high temperatures for a longer time when the energy density is high. The expansion of thermal cracking in the molten metal will be aided by this. Consequently, by decreasing the heat input during laser welding, thermal cracking can be reduced.

### 3.3. Effect of Pulsed Laser Welding Process Parameters on Cracking

Pulsed laser thermal shock properties lead to micro-cracks in the welded joint, and the heat treatment method of the metal material and its own organization are also factors in the formation of cracks. For high temperature alloys, solidification cracks are mainly associated with low melting point materials, such as the laves phase, and when the melt pool solidifies faster, the laves phase precipitates less and cracks are less frequent or less damaging. Melt pool solidification rate and welding parameters, therefore, effect the degree of microcracking in the welded joint. Experiments with different welding parameters were conducted to study the effect of pulsed laser welding parameters on the degree of cracking in the welded joint. Figure 8 and Figure 9 show the results of different parameters after welding the surface of the welded joint and the length of the crack in the welded joint, and the resulting statistical chart.

Using a slowly rising and falling waveform for the images shown in Figure 8a and a rectangular waveform for the pulsed laser welded joint shown in Figure 8b, the crack length statistics were compiled that are shown in Figure 9a. When Figure 8a,b are compared, it is clear that the slowly rising and dropping waveform helps to lessen the amount of cracking in the welded joint. As seen in the image, there are fractures in the welded joint that extend from within the welded joint to the heat-affected zone. These cracks are often seen in the heat-affected zone and inside the welded joint. Rectangular wave welding causes a larger melt pool depth and more severe fracture damage to the welded connection. Low melting point materials, such as Laves phase, are the principal culprits for solidification cracking. As the melt pool solidifies more quickly, Laves phase precipitates less, causing less damage or cracking. When employing a slow rise and slow fall waveform as opposed to a rectangular waveform with the same parameters, the heat input per laser pulse is smaller, the melt pool solidifies quicker, there is less precipitation, and there is less cracking in the welding spot.

Figure 8a,b displays the various laser peak powers, and the crack length statistics are shown in Figure 9a. In the case of the same average laser power, the peak laser power plays a major role in the melt pool width and melting depth. When the peak power increases, the welding pool melting depth steadily increases. The depth of the melt pool’s melting increases noticeably when the laser’s peak power approaches 2000 watts. When the peak power is higher, the laser beam capacity is adequate to create a keyhole on the surface of the metal material, which causes a significant increase in the amount of laser energy absorbed by the melt pool. When the laser power is higher, the deep width of the welded joint is relatively wide because the molten pool’s slow rate of heat dissipation after the conclusion of a laser pulse preheats the subsequent pulse of welding if the laser frequency is high. When the laser peak power is high, the molten pool solidification speed is slow, and the solidification process is prone to cracking; therefore, reducing the laser power can reduce the cracks in the welded joint. When the peak laser power is low, the possibility of cracks on the fusion line is higher, and the cracks extend from the fusion line to the heat-affected zone. When the welding parameters make the molten pool size smaller, the center of the welded joint is more likely to produce cracks, and the crack size is larger. Moreover, by optimizing the welding parameters, it is not possible to suppress the cracks produced by welding 718 alloy, but only to reduce the number of cracks and decrease the degree of crack damage.

Experiments of welded joints with a peak power of 2000 w but different pulse durations are shown in Figure 8c, and the crack length statistics as shown in Figure 9b. According to Figure 8c, we observe that by increasing the pulse duration, the melt width becomes larger and the melt depth becomes smaller; moreover, as the pulse duration increases, the crack size in the welded joint becomes smaller and the number of cracks decreases. When the pulse duration is longer, the laser heating pool time becomes longer and the melt pool solidification becomes slower, so that the melt pool width is large; under the same larger pulse time interval, the melt width is large, and the heat in the melt pool is dissipated quickly, resulting in the melt depth of the melt pool becoming smaller. A long pulse duration leads to a slower molten pool solidification speed and less cracks in the welded joint, therefore providing a method to reduce crack damage.

The welded joints of different pulse frequency welding experiments are shown in Figure 8d, and the crack length is shown in Figure 9c. We observe from Figure 8d that the melt width of the molten pool becomes larger and the melt depth becomes larger for larger laser pulse frequencies; in addition, the crack size in the welded joint becomes larger and the number of cracks decreases as the laser pulse frequency becomes larger. When the frequency is larger, the pulse interval becomes shorter, so that more heat remains in the melt pool into the next laser pulse action; therefore, the melt pool depth is larger, the melt pool heat dissipation is slow, and solidification becomes slower, increasing the degree of crack damage. When the melting depth is larger and the pulse interval is shorter, the liquid melt pool content in the melt pool paste area becomes larger when the next laser pulse occurs, so that the melt width becomes larger.

## 4. Conclusions

In this paper, spot welding experiments with different waveforms, peak powers, frequencies, and pulse widths were conducted to study the cracks generated by pulsed laser thermal shock, and based on the experimental results and analysis, the following conclusions can be drawn.

The formation of cracks in the welded joint is related to thermal shock, where cavitation is formed in the molten pool under the effect of thermal shock, which becomes the source of cracks.The Nb and C contents of cracks within the welded joints were analyzed by EDS. Inconel 718 alloy in the rolled state, during rapid solidification, is enriched with a large amount of Nb elements between the dendrites and grain boundaries, forming a liquid film with Laves phase.The use of a slow rise and slow fall waveform is conducive to reducing crack damage, as the melt depth of the melt pool becomes larger; reducing the peak laser power to 1000 w is conducive to reducing the crack in the welded joint; by increasing the pulse duration to 20 ms, the melt depth of the melt pool becomes smaller, and the degree of crack damage is reduced; finally, reducing the pulse frequency to 10 Hz can reduce the degree of crack damage. Changing the laser parameters can reduce cracks in the welded joints, and in addition, the findings have more important implications for future studies of crack sensitivity in the heat-affected zone induced by the thermal shock of pulsed laser spot welding.

## Figures and Tables

**Figure 1 materials-16-03775-f001:**
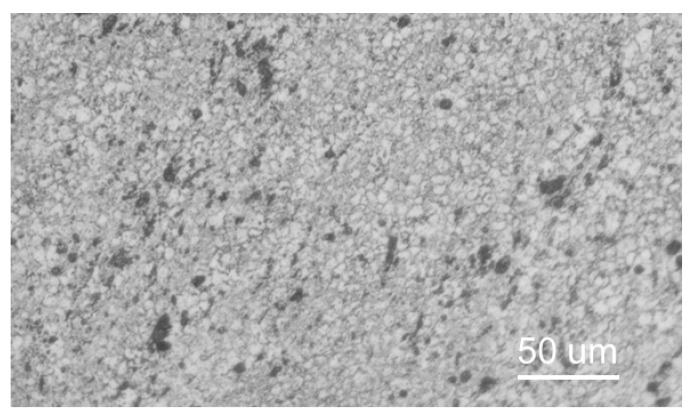
Organization of Inconel 718 alloy base material.

**Figure 2 materials-16-03775-f002:**
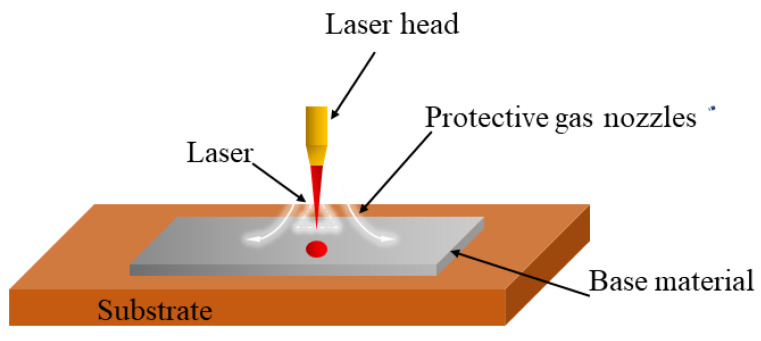
Diagram of spot-welding experimental equipment.

**Figure 3 materials-16-03775-f003:**
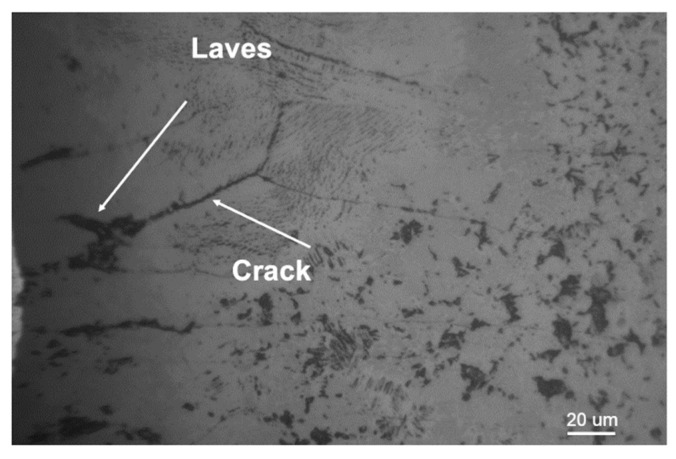
Cracks in solder joints and surrounding precipitates.

**Figure 4 materials-16-03775-f004:**
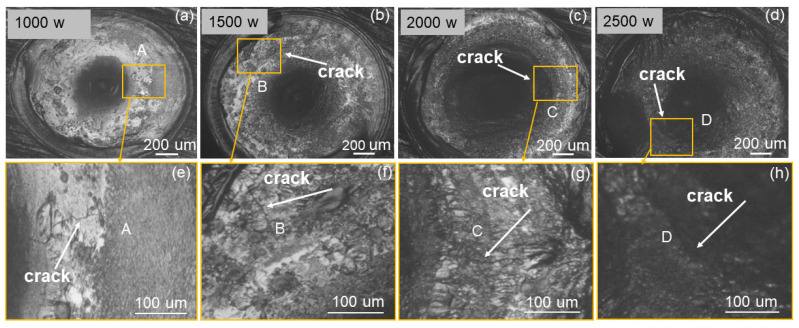
(**a**–**d**) Surface of welded joints after welding at different peak powers, (**e**–**h**) are enlarged images of A–D with cracks in (**a**–**d**) respectively.

**Figure 5 materials-16-03775-f005:**
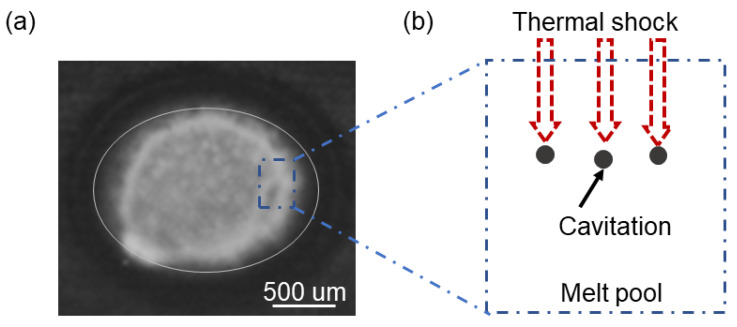
(**a**) Image of the melt pool recorded by high-speed camera, and (**b**) schematic diagram of the pulsed laser thermal shock acting to produce cavities in the paste region.

**Figure 6 materials-16-03775-f006:**
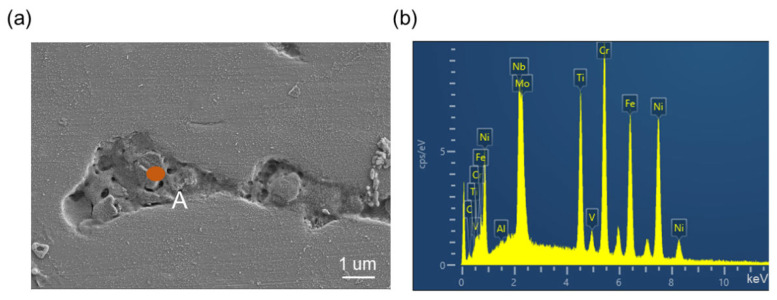
(**a**) Laves phase precipitates near cracks in solder joints, and (**b**) EDS phase spectrum of Laves phase A region.

**Figure 7 materials-16-03775-f007:**
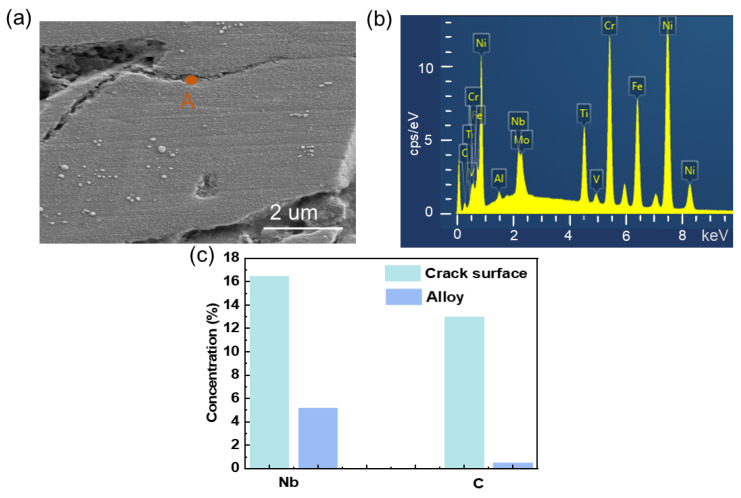
(**a**) SEM images of cracks in solder joints, (**b**) EDS spectra of region A of the crack cross section and (**c**) elemental content analysis of alloy (blue) and cracked surface (green) (from EDS).

**Figure 8 materials-16-03775-f008:**
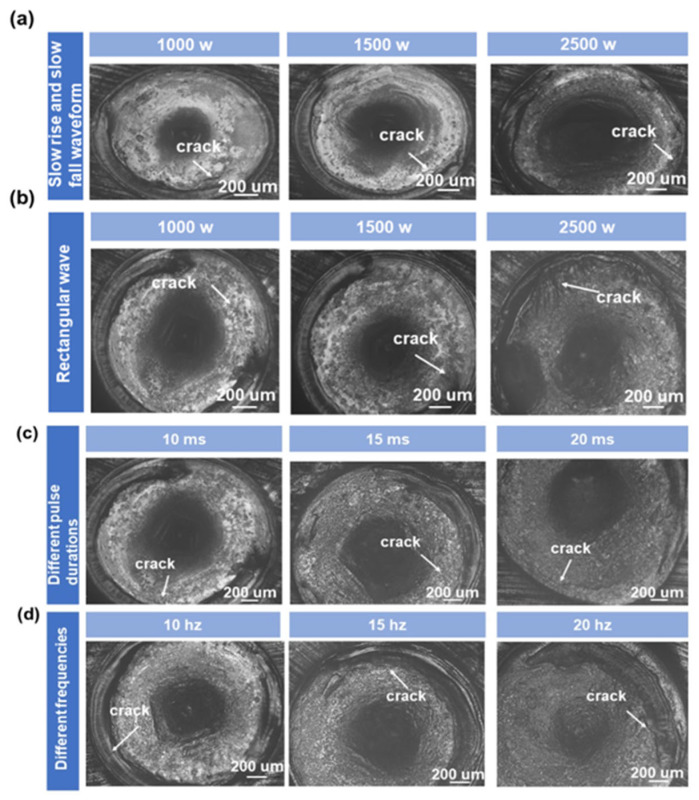
(**a**) using slow rise and slow fall waveform at different peak power, (**b**) using rectangular wave at different peak power, (**c**) different pulse duration, (**d**) different laser pulse frequency obtained solder joint surface.

**Figure 9 materials-16-03775-f009:**
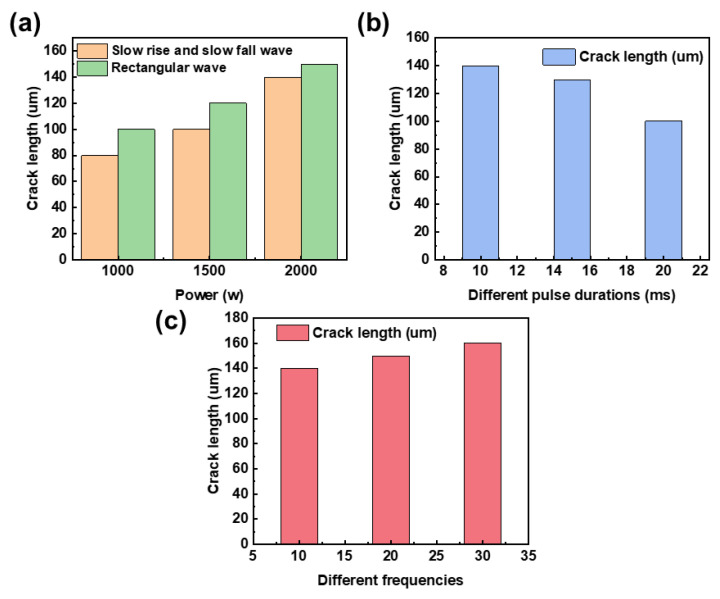
(**a**) Crack lengths in welded joints with different waveforms with different peak powers, (**b**) different pulse durations, and (**c**) different laser frequencies.

**Table 1 materials-16-03775-t001:** Chemical composition of Inconel 718 alloy.

Elements	Chemical Components (wt%)
Ni	53.21
Mo	3.12
Cr	18.99
Ti	1.05
Cu	0.15
Nb	4.86
Al	0.49
C	0.06
Si	0.18
Fe	Bal.

**Table 2 materials-16-03775-t002:** Mechanical properties of Inconel 718 alloy at 720 °C.

Material	Tensile Stress (MPa)	Proof Stress (MPa)	Hardness (HRC)	γ′ Strengthening (MPa)	γ″ Strengthening (MPa)
Inconel 718	1367.76	1022.3	38.55	242.31	481.28

## Data Availability

Not applicable.

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
