# Peer review of "Pulsed Laser Spot Welding Thermal-Shock-Induced Microcracking of Inconel 718 Thin Sheet Alloy"

_materials, 2023, doi:10.3390/ma16103775_

Round 1

Reviewer 1 Report

In this work, the authors studied the Inconel 718 alloy as the welding substrate and pulsed laser spot welding to conduct welding tests with various parameters. studying the pressures produced by thermal shock to investigate crack damage from pulsed laser thermal shock. The research appears to be efficiently done and appropriately reported, however, the standard of English is acceptable but only needs a few corrections. Nevertheless, some questions and corrections must be answered to improve and complete the manuscript.

Abstract section: The abstract must be improved; I suggest to authors follow these rules:

A. One or two sentences on BACKGROUND

B. Two or three sentences on METHODS

C. Less than two sentences on RESULTS

D. One sentence on CONCLUSIONS

Introduction section: In this section, the authors don’t indicate the novelty of their work. what is the innovation of your work when compared with the other researchers? The "Knowledge gap to be filled"? In this introduction, the authors must describe or indicate the work that will be done to test their "hypothesis". On the other hand, this study was based on the identification of structural damage (cracks caused by welding), however, the authors make no reference to techniques to measure and locate the damage. I suggest that you complete your theoretical analysis based on the study described in the reference with doi 10.3233/SAV-2012-0692.

Experiment section: The authors did not refer to what kind of laser they used: CO2? Nd/YAG? …. The power density was enough for penetration welding (keyhole) or conduction welding?

The numbering of Figures, in most  cases, is completely wrong. The authors must verify and correct them.

During the text, there are a lot of mistakes in referring to the figure numbers. Some examples_ line 59 (Figure 3(a)); lines 200, 201, 202 (Figures 4(a), 4(b), 4(c)); lines 230, 233, 244, 264, …

Lines 76, 113, 114. Please, very the units, I think it is the micrometer (20, 20-40, 100).

The standard of English is acceptable but only needs a few corrections.

Author Response

Response to reviewer 1 Comments

Thank you very much for the comments you gave, they were very appropriate and useful, and some changes have been made to the English grammar of the manuscript. I will respond to the comments and questions one by one below.

Point 1: Abstract section: The abstract must be improved; I suggest to authors follow these rules:

  1. One or two sentences on BACKGROUND
  2. Two or three sentences on METHODS
  3. Less than two sentences on RESULTS
  4. One sentence on CONCLUSIONS

Response 1:

Changes have been made to the abstract section, corresponding to lines 11 to 25 in the text.

This paper investigates the change in solidification microcrack susceptibility under the influence of thermal shock-induced effects for pulsed laser spot welding molten pools with different waveforms, powers, frequencies, and pulse widths. During the welding process the molten pool under the effect of thermal shock temperature changes sharply, triggering pressure waves, creating cavities in the molten pool paste area and forming crack sources during solidification. The microstructure near the crack was analyzed using SEM (scanning electron microscope) and EDS (electronic differential system), and it was found that bias precipitation occurred during the rapid solidification of the melt pool, and a large amount of Nb elements were enriched in the interdendrites and grain boundaries, which eventually formed a liquid film with a low melting point Laves phase. When cavities appear in the liquid film, the chance of crack source formation is further increased. Using a slow rise and slow fall waveform is good for reducing cracks; reducing the peak laser power to 1000 w is good for reducing cracks in the solder joint; increasing the pulse width to 20 ms reduces the degree of crack damage; reducing the pulse frequency to 10 hz reduces the degree of crack damage.

Point 2: Introduction section: In this section, the authors don’t indicate the novelty of their work. what is the innovation of your work when compared with the other researchers? The "Knowledge gap to be filled"? In this introduction, the authors must describe or indicate the work that will be done to test their "hypothesis". On the other hand, this study was based on the identification of structural damage (cracks caused by welding), however, the authors make no reference to techniques to measure and locate the damage. I suggest that you complete your theoretical analysis based on the study described in the reference with doi 10.3233/SAV-2012-0692.

Response 2:

  • The main contents of the manuscript are redescribed in the introduction, lines 84-93, and the innovations of our work are added.

The issue of thermal shock welding has received a great deal of attention, and the heat-affected zone cracking induced by electron beam thermal shock has been studied, as well as the exploration of thermal excitation caused by pulsed laser welding. In order to further study the pulsed laser spot welding melt pool, micro-cracking induced by thermal shock, this paper chose the rolled state Inconel 718 alloy sheet as the weld substrate and carried out spot welding experiments with different parameters. By analyzing and calculating the melt pool pressure, the pulsed laser thermal shock effect is studied and the crack formation mechanism is elaborated. Analysis of cracking and precipitation phase composition in welded joints using SEM and EDS to explore crack damage from pulsed laser thermal shock.

  • In addition, in the introduction section, lines 79-83 add other scholars on the measurement methods and techniques of structural damage and welding crack damage

In addition, the measurement and detection of crack damage plays an important role in the study of crack damage. Lopes [22] proposed several interferometric techniques and their applications in structural damage identification. Amirhossein Mashuriazar [23] used ImageJ to measure the area fraction, quantity and total length of cracks in the heat affected zone.

  • The means of crack measurement is added in the experimental section, lines 129-131.

In order to study the variation of the size and number of cracks in welded joints with different laser parameters, the images of the welded joint surfaces were processed using imagej to count the size and number of cracks in the welding spot.

Point 3: Experiment section: The authors did not refer to what kind of laser they used: CO2? Nd/YAG? …. The power density was enough for penetration welding (keyhole) or conduction welding?

Response 3:

We used Nd:YAG (neodymium-doped yttrium aluminum garnet) laser and chose Inconel 718 alloy sheet for pulsed laser conduction spot welding, which has been modified in the experimental section, lines 120-123.

The welding experiments are mainly pulsed laser spot welding using Nd:YAG (neodymium-doped yttrium aluminum garnet) laser with a wavelength of 1064 nm. Pulsed laser conduction spot welding with different power, frequency and pulse width was performed under two waveforms.

Point 4: The numbering of Figures, in most  cases, is completely wrong. The authors must verify and correct them. During the text, there are a lot of mistakes in referring to the figure numbers. Some examples_ line 59 (Figure 3(a)); lines 200, 201, 202 (Figures 4(a), 4(b), 4(c)); lines 230, 233, 244, 264, …

Response 4:

The figures have been carefully checked and corrected, as well as the numbering of the figure references in the text.

Point 5: Lines 76, 113, 114. Please, very the units, I think it is the micrometer (20, 20-40, 100).

Response 5:

The units are microns and have been revised in lines 140 and 142 of the text.

Reviewer 2 Report

The paper entitled “Inconel 718 pulsed laser welding thermal shock induced microcrack" presents the analysis of the effect of various waveforms, peak powers, frequencies, 15 and pulse durations on laves formation in Inconel 718. From my point of view, the topic is of great interest. But in general, need some reviews to be done:

·        The abstract provides a good overview of the research and its findings. However, it could benefit from a clearer statement of the research question or objective at the beginning. Additionally, it might be helpful to include a brief explanation of the significance or implications of the research.

·        In the introduction references more actual references could be address. As examples:

o   https://doi.org/10.1016/j.jmst.2022.02.015

o   https://doi.org/10.3390/ma12132159

o   https://doi.org/10.30657/pea.2017.15.04

·        The figures are in general very visually appealing, but they are a bit blurred and with small labels (e.g. Fig. 4, the labels of the axes are not well readable).

·        In general the structure of the paper is unacceptable as this, Figures appear without being cited (e.g. Fig. 1).

·        Figures referenced after appearing (e.g. Fig. 2).

·        Figure 2 is very confusing, have a better picture of the experimental set up, maybe add a schematic of the device.

·        There must be a space between the number and the designation of the units. (e.g. 200 mm instead of 200mm)

·        Figure 5 literally shows nothing and is not scaled.

·        Jumping from Figure 4 to 6 to 9 ¿?

·        Explain the acronyms the first time they are used (e.g. Nd:YAG (neodymium-doped yttrium aluminum garnet)).

·        The conclusions might include a brief discussion of the broader implications of the research.

·        It might be useful to include a brief statement about future research directions or potential areas for further investigation.

·        References are not correctly addressed on the reference section.

The text could be clearer and better organised.

Author Response

Response to reviewer 2 Comments

Thank you very much for the comments you gave, they are very appropriate and useful, and I will respond to the comments and questions one by one below.

Point 1: In the introduction references more actual references could be address. As examples:

https://doi.org/10.1016/j.jmst.2022.02.015

https://doi.org/10.3390/ma12132159

https://doi.org/10.30657/pea.2017.15.04

Response 1:

The suggestions you gave were very useful and the summary section has been revised

This paper investigates the change in solidification microcrack susceptibility under the influence of thermal shock-induced effects for pulsed laser spot welding molten pools with different waveforms, powers, frequencies, and pulse widths. During the welding process the molten pool under the effect of thermal shock temperature changes sharply, triggering pressure waves, creating cavities in the molten pool paste area and forming crack sources during solidification. The microstructure near the crack was analyzed using SEM (scanning electron microscope) and EDS (electronic differential system), and it was found that bias precipitation occurred during the rapid solidification of the melt pool, and a large amount of Nb elements were enriched in the interdendrites and grain boundaries, which eventually formed a liquid film with a low melting point Laves phase. When cavities appear in the liquid film, the chance of crack source formation is further increased. Using a slow rise and slow fall waveform is good for reducing cracks; reducing the peak laser power to 1000 w is good for reducing cracks in the solder joint; increasing the pulse width to 20 ms reduces the degree of crack damage; reducing the pulse frequency to 10 hz reduces the degree of crack damage.

In addition, in the introduction, lines 42-48 and 57-61 add some of the contents of these three given documents

Inconel 718 alloy is used in a wide range of applications, such as chemical, petrochemical, energy and aerospace. Txomin Ostra's research concluded that the use of laser metal deposition (LMD) allows the manufacture of near-net shape products, implying significant savings in terms of materials and costs in the manufacture of high-performance components for the aerospace industry [7,8]

Janaki Ram et al [12] used welding current pulsation in GTAW of Inconel 718 and observed dendrite refinement in FZ. Sivaprasad et al [13] implemented arc oscillation in the GTAW of Inconel 718 alloy and observed that dendrite refinement in the FZ led to the development of finer Laves phases in the FZ.

Point 2: The figures are in general very visually appealing, but they are a bit blurred and with small labels (e.g. Fig. 4, the labels of the axes are not well readable).

Response 2:

Modifications have been made to Figure 4, as well as to the labels of the axes in Figure 7.

Figure 4. (a)-(d) Surface of welded joints after welding at different peak powers; (e)-(h) corresponding to cracks in welded joints at different peak powers.

Figure 7 (a) SEM images of cracks in solder joints, (b) EDS spectra of cracks. Surface Nb, and (C) elemental content analysis of alloy (blue) and cracked surface (green) (from EDS).

Point 3: In general the structure of the paper is unacceptable as this, Figures appear without being cited (e.g. Fig. 1).

Response 3:

A reference to Figure 1 has been added to line 98 in the manuscript.

The microstructure of the base material is shown in Figure 1,

Point 4: Figures referenced after appearing (e.g. Fig. 2).

Response 4:

The numbering of figures and the numbering of figures cited in the text have been carefully checked and corrected.

Point 5: Figure 2 is very confusing, have a better picture of the experimental set up, maybe add a schematic of the device.

Response 5: Changes were made to Figure 2 in the manuscript.

Point 6: There must be a space between the number and the designation of the units. (e.g. 200 mm instead of 200mm)

Response 6:

Modified in line 96 of the text, and this issue was carefully checked and corrected in the manuscript.

Point 7: Figure 5 literally shows nothing and is not scaled.

Response 7: The figure has been modified in the manuscript.

Figure 5. (a) image of the melt pool recorded by high-speed camera, (b) schematic diagram of the pulsed laser thermal shock acting to produce cavities in the paste region.

Point 8: Jumping from Figure 4 to 6 to 9 ?

Response 8:

The numbering of figures and the numbering of figures cited in the text have been carefully checked and corrected.

Point 9: Explain the acronyms the first time they are used (e.g. Nd:YAG (neodymium-doped yttrium aluminum garnet)).

Response 9:

SEM (Scanning Electron Microscope), EDS (Electron Differential System) explained in line 17, Nd: YAG (Neodymium Doped Yttrium Aluminum Garnet), ABB (Asea Brown Boveri) explained in line 30.

Point·10: The conclusions might include a brief discussion of the broader implications of the research.

Response 10:

The conclusion has been revised in lines 317-332 of the text.

In this paper, spot welding experiments with different waveforms, peak powers, frequencies, and pulse widths were conducted to study the cracks generated by pulsed laser thermal shock, and based on the experimental results and analysis, the following conclusions can be drawn.

1.The formation of cracks in the welded joint is related to thermal shock, where cavitation is formed in the molten pool under the effect of thermal shock, which becomes the source of cracks.

2.The Nb and C contents of cracks within the welded joints were analyzed by EDS. Inconel 718 alloy in the rolled state, during rapid solidification, is enriched with a large amount of Nb elements between the dendrites and grain boundaries, forming a liquid film with Laves phase.

3.The use of slow rise and slow fall waveform is conducive to reducing crack damage, the melt depth of the melt pool becomes larger; reduce the peak laser power to 1000w, conducive to reducing the crack in the welded joint; increase the pulse width to 20ms, the melt depth of the melt pool becomes smaller, the degree of crack damage is reduced; reduce the pulse frequency to 10Hz, can reduce the degree of crack damage.

Point·11: It might be useful to include a brief statement about future research directions or potential areas for further investigation.

Response 11:

In this paper, microcracks in solder joints induced by pulsed laser thermal shock are investigated, which may be useful for studying cracks in the heat-affected zone induced by pulsed laser thermal shock, which can next be investigated by the heat-affected zone cracks induced by pulsed laser thermal shock action. In addition, the innovation and significance of this study are explained in the introduction section, lines 84-93.

The issue of thermal shock welding has received a great deal of attention, and the heat-affected zone cracking induced by electron beam thermal shock has been studied, as well as the exploration of thermal excitation caused by pulsed laser welding. In order to further study the pulsed laser spot welding melt pool, micro-cracking induced by thermal shock, this paper chose the rolled state Inconel 718 alloy sheet as the weld substrate and carried out spot welding experiments with different parameters. By analyzing and calculating the melt pool pressure, the pulsed laser thermal shock effect is studied and the crack formation mechanism is elaborated. Analysis of cracking and precipitation phase composition in welded joints using SEM and EDS to explore crack damage from pulsed laser thermal shock.

  • Point 12: References are not correctly addressed on the reference section.

Some references have been replaced and some additional references have been added. In addition, lines 42-83 of the introduction section of the text have been revised according to the references.

  1. Ostra, T.; Alonso, U.; Veiga, F.; Ortiz, M.; Ramiro, P.; Alberdi, A. Analysis of the Machining Process of Inconel 718 Parts Manufactured by Laser Metal Deposition. Materials 2019, 12, 2159, doi:10.3390/ma12132159.
  2. Jambor, M.; Bokůvka, O.; Nový, F.; Trško, L.; Belan, J. Phase Transformations in Nickel base Superalloy Inconel 718 during Cyclic Loading at High Temperature. Production Engineering Archives 2017, 15, 15–18, doi:10.30657/pea.2017.15.04.
  3. Sui, S.; Li, H.; Li, Z.; Zhao, X.; Ma, L.; Chen, J. Introduction of a New Method for Regulating Laves Phases in Inconel 718 Superalloy during a Laser-Repairing Process. Engineering 2022, 16, 239–246, doi:10.1016/j.eng.2021.08.030.
  4. Radhakrishna, C.; Rao, K.P. The formation and control of Laves phase in superalloy 718 welds. Journal of Materials Science 1997, 32, 1977–1984, doi:10.1023/A:1018541915113.
  5. Fu, J.; Li, H.; Song, X.; Fu, M.W. Multi-scale defects in powder-based additively manufactured metals and alloys. Journal of Materials Science & Technology 2022, 122, 165–199, doi:10.1016/j.jmst.2022.02.015.
  6. Janaki Ram, G.D.; Venugopal Reddy, A.; Prasad Rao, K.; Madhusudhan Reddy, G. Control of Laves phase in Inconel 718 GTA welds with current pulsing. Science and Technology of Welding and Joining 2004, 9, 390–398, doi:10.1179/136217104225021788.
  7. Sivaprasad, K.; Ganesh Sundara Raman, S.; Mastanaiah, P.; Madhusudhan Reddy, G. Influence of magnetic arc oscillation and current pulsing on microstructure and high temperature tensile strength of alloy 718 TIG weldments. Materials Science and Engineering: A 2006, 428, 327–331, doi:10.1016/j.msea.2006.05.046.
  8. Lingenfelter, A. Welding of Inconel Alloy 718: A Historical Overview. The Minerals,Metals&Materials Society, 1989, pp 673–683, doi:10.7449/1989/Superalloys_1989_673_683.
  9. Sujan, G.K.; Gazder, A.A.; Awannegbe, E.; Li, H.; Pan, Z.; Liang, D.; Alam, N. Hot Deformation Behavior and Microstructural Evolution of Wire-Arc Additively Fabricated Inconel 718 Superalloy. Metall Mater Trans A 2023, 54, 226–240, doi:10.1007/s11661-022-06863-3.
  10. Ye, X.; Hua, X.; Wang, M.; Lou, S. Controlling hot cracking in Ni-based Inconel-718 superalloy cast sheets during tungsten inert gas welding. Journal of Materials Processing Technology 2015, 222, 381–390, doi:10.1016/j.jmatprotec.2015.03.031.
  11. Sonar, T.; Balasubramanian, V.; Malarvizhi, S.; Venkateswaran, T.; Sivakumar, D. An overview on welding of Inconel 718 alloy - Effect of welding processes on microstructural evolution and mechanical properties of joints. Materials Characterization 2021, 174, 110997, doi:10.1016/j.matchar.2021.110997.
  12. Lin, J.; Wang, X.; Lei, Y.; Wei, R.; Guo, F. Study on hot cracking in laser welded joints of inconel 718 alloy foils. Journal of Manufacturing Processes 2021, 64, 1024–1035, doi:10.1016/j.jmapro.2021.02.002.
  13. Low, Z.K.; Chaise, T.; Bardel, D.; Cazottes, S.; Chaudet, P.; Perez, M.; Nelias, D. A novel approach to investigate delta phase precipitation in cold-rolled 718 alloys. Acta Materialia 2018, 156, 31–42, doi:10.1016/j.actamat.2018.06.005.
  14. Han, D.-W.; Yu, L.-X.; Liu, F.; Zhang, B.; Sun, W.-R. Effect of Heat Treatment on the Microstructure and Mechanical Properties of the Modified 718 Alloy. Acta Metall. Sin. (Engl. Lett.) 2018, 31, 1224–1232, doi:10.1007/s40195-018-0790-9.
  15. Keivanloo, A.; Naffakh-Moosavy, H.; Miresmaeili, R. The effect of pulsed laser welding on hot cracking susceptible region size and weld pool internal geometry of Inconel 718: Numerical and experimental approaches. CIRP Journal of Manufacturing Science and Technology 2021, 35, 787–794, doi:10.1016/j.cirpj.2021.09.001.
  16. Lopes, H.; Ribeiro, J.; Araújo Dos Santos, J.V. Interferometric Techniques in Structural Damage Identification. Shock and Vibration 2012, 19, 835–844, doi:10.1155/2012/471510.
  17. Mashhuriazar, A.; Omidvar, H.; Gur, C.H.; Sajuri, Z. Effect of Welding Parameters on the Liquation Cracking Behavior of High-Chromium Ni-Based Superalloy. J. of Materi Eng and Perform 2020, 29, 7843–7852, doi:10.1007/s11665-020-05292-w.
  18. Wang, N.; Mokadem, S.; Rappaz, M.; Kurz, W. Solidification cracking of superalloy single- and bi-crystals. Acta Materialia 2004, 52, 3173–3182, doi:10.1016/j.actamat.2004.03.047.

Reviewer 3 Report

The paper needs to be rewritten. Please cross-check the figure and table no. citation in the text. The found many errors in terms of citations that lost my interest in reading the manuscript. 

1. Table no. 2 is missing which is the mechanical properties of the work materials. 

2. Figure 1. is not cited in the text.

3. The spot diameter of the weld is written as 600 m, need to check.

4. Fig. 2 is written twice in the text.

5. Figure 4. Picture quality needs to improve.

Please check the manuscript thoroughly and resubmit it. 

Need to improve

Author Response

Response to reviewer 3 Comments

We apologize for the unwarranted errors in the manuscript and the bad experience it has given you. We have checked the manuscript thoroughly and answered all your questions and suggestions.

Point 1: Table no. 2 is missing which is the mechanical properties of the work materials. 

Response 1:

The mechanical properties of Inconel 718 alloy with chemical composition content of Table 1 were calculated using JMatpro to obtain Table 2 at 720°C and were added to the manuscript.

Table 2. Mechanical properties of Inconel 718 alloy at 720℃.

Material

tensile stress

(MPa)

Proof Stress

(MPa)

Hardness

(HRC)

strengthening

(MPa)

strengthening

(MPa)

Inconel 718

1367.76

1022.3

38.55

242.31

481.28

Point 2: Figure 1. is not cited in the text.

Response 2: A reference to Figure 1 has been added to line 98 in the manuscript.

The microstructure of the base material is shown in Figure 1,

Point 3: The spot diameter of the weld is written as 600 m, need to check.

Response 3: Revised line 117 of the manuscript.

Point 4: Fig. 2 is written twice in the text.

Response 4: The figure numbers in the manuscript and the numbering of figures cited in the text have been double-checked and corrected.

Point 5: Figure 4. Picture quality needs to improve.

Please check the manuscript thoroughly and resubmit it. 

Response 5: Changes were made to Figure 4 in the manuscript.

Figure 4. The surface of the solder joint after soldering at different peak powers.

Round 2

Reviewer 2 Report

The paper is now of quality to be publish

Author Response

Reply to the Review Report

The references in the manuscript have been carefully checked, and the manuscript has been revised in response to the referee's comments, and each of them has been answered.

Point 1: (Page 2, Line 63) Please, replace: “structural liquefaction of Nb/C, Laves at grain” with “structural liquefaction of Nb/C and Laves phases at grain”

Reponse1: Revisions have been made to lines 60-61 in the manuscript.
Point 2: (Page 2, Lines 43-46) Please, remove the sentence: “Txomin Ostra's research concluded that the use of laser 43 metal deposition (LMD) allows the manufacture of near-net shape products, implying significant savings in terms of materials and costs in the manufacture of high- performance components for the aerospace industry [7,8].” These references are on additive manufacturing, and the topic is on laser welding.

Reponse2: Reference [7,8] has been removed from the manuscript and the citations have been deleted from the introduction.
Point 3: (Page 2, Lines 80-81) Please, remove the sentence: “Lopes [22] proposed 80 several interferometric techniques and their applications in structural damage identification.” This statement is not related to the aim of the paper. This reference is on structural integrity, a topic nonrelated with the main aim of the manuscript.

Reponse3: References have been removed from the manuscript [22], and relevant citations have been removed from the introduction.

Reviewer 3 Report

Please add some future work in connection with the current work in the conclusions section. 

Author Response

Point: Please add some future work in connection with the current work in the conclusions section.

Response: Added to lines 326 - 328 in the manuscript.

 Changing the laser parameters can reduce cracks in the welded joints, and in addition the findings have more important implications for future studies of crack sensitivity in the heat-affected zone induced by thermal shock of pulsed laser spot welding.
